

# Do the regular annual extreme water level changes affect the seasonal appearance of *Anabaena* in Poyang Lake?

Kuimei Qian[1], Martin Dokulil[2] and Yuwei Chen[3]

[1] College of Environmental Engineering, Xuzhou University of Technology; Jiangsu Laboratory of Pollution Control and Resources Reuse, Xuzhou, China
[2] Research Department for Limnology, Mondsee, University of Innsbruck, Monsee, Austria
[3] Nanchang Institute of Technology, Nanchang, China

## ABSTRACT

**Background:** Poyang Lake is an ecosystem experiencing annual variations in water level of up to 14 m. Water level changes were 8.03 and 11.22 m, respectively, in the years 2013 and 2014. The biomass and heterocyst frequency of *Anabaena* increased in the summers of recent years.

**Methods:** A weekly to bi-weekly monitoring from June to November 2013 and 2014 was set up to explain the variations of *Anabaena* appearance in different phases of the water level.

**Results:** *Anabaena* was present in the lake throughout the year. The average relative biomass of *Anabaena* in the present study was over 40%, being most abundant in summer. The average heterocyst frequency was 0.23% in 2013 and 0.76% in 2014. Correlation analysis indicated a positive trend of *Anabaena* biomass with water temperature and water level and a negative one with total nitrogen (TN), which is the reason for the increase of heterocyst frequency in 2013 and 2014. Heterocyst frequency of *Anabaena* was positively correlated with water temperature, water level and $PO_4$-P, and negatively with dissolved inorganic nitrogen (DIN/DIP), $NO_3$-N and TN. Moreover, water temperature and DIN/DIP were significantly correlated with water level, indicating that water level changes have a direct effect on *Anabaena* and heterocyst formation in Poyang Lake.

**Conclusions:** The results of this study support the hypothesis that increasing biomass and heterocyst formation of *Anabaena* can be primarily caused by seasonal changes of the water level in Poyang Lake.

Corresponding author
Kuimei Qian,
qiankuimei@xzit.edu.cn

## INTRODUCTION

As a key variable in hydrology, changes in water level have significant effects on lake ecology and management affecting environmental factors such as turbidity and transparency (*Coops, Beklioglu & Crisman, 2003*; *Domitrovic, 2003*; *Lopes, Bicudo & Ferragut, 2005*; *Mihaljević & Stević, 2011*; *O'Farrell et al., 2011*). Such water level change is usually moderate, multi-annual or annual (*Zohary & Ostrovsky, 2011*; *Casali et al., 2011*;

*Da Costa, Attayde & Becker, 2016*; *Fuentes & Petrucio, 2015*), Poyang Lake experiences a pulse by up to 14 m once every year (*Zhang et al., 2014*). The water level change has an important role in the transporting of organisms and nutrients throughout the lake in the direction of the water flow from south to north. Environmental changes are largely responsible for temporal patterns in phytoplankton (*Huszar et al., 1998*) which in turn might be used as an ecological tool to analyze short-term responses (*Reynolds, 2002*; *Rodrigues et al., 2002*). Cyanobacterial $N_2$-fixation can be of ecological importance in nitrogen-deficient water (*Karl et al., 2002*) because fixing atmospheric N provides combined nitrogen to the pelagic ecosystem and hence supports new planktonic production (*Capone & Carpenter, 1982*).

Each habitat in the floodplain has different factors that influence the structure and dynamic of aquatic communities. Poyang Lake can be classified as highly eutrophic, due to the excessive inputs from agriculture, industry and waste discharges during intensive economic activities and growing human population in recent decades (*Yang et al., 2015*). Nutrients in Poyang Lake caused the proliferation of algae (*Deng et al., 2011*; *Zhen et al., 2011*; *Liu et al., 2016a*). Total phytoplankton biomass was high in autumn of the years 2010–2013, associated with cyanobacterial blooms in some regions of the lake in these years (*Qian et al., 2016b*). Assemblages of *Anabaena spp. and Microcystis spp.* migrated to the lake surface in some lentic regions and were visible even at the surface in the outlet channel of Poyang Lake. Therefore, not only the buoyant Cyanobacteria but also the nutrients flow to this area along with the water current. There is a clear shift of the dominant Cyanobacteria species from *Microcystis spp.* to *Anabaena spp.* in the late summer and early autumn. *Anabaena,* as a nitrogen-fixing species, formed heterocyst in this time.

The aim of the study was to find out which factors affect the seasonal appearance of *Anabaena* in Poyang Lake. We hypothesize that the transition in water level is responsible for shifts in species domination within the Cyanobacterial assemblage from *Microcystis* to *Anabaena* and back. We further hypothesize that increasing biomass and heterocyst formation of *Anabaena* can be primarily caused by seasonal changes in water level. Such fluctuations alter the physical and chemical parameters of a system, such as Poyang Lake.

## MATERIALS AND METHODS

### Study area

Poyang Lake (28°22′N–29°45′N, 115°47′E–116°45′E) is located in northern Jiangxi Province. Water in the lake flows from south to north to discharge into the Yangtze River through a narrow outlet at Hukou (Fig. 1). The water level and its annual change are determined by discharges from the five sub-tributaries and the climatic variability in the region. Precipitation increases rapidly from March to August and decreases after September. In response to the annual cycle of precipitation, Poyang Lake has four different phases: low water level phase, increasing water level phase, high water level phase, and decreasing water level phase. The water level will increase to higher than 14 m above sea level (Wu Song datum) at Xingzi in the high water level phase, with the maximum between

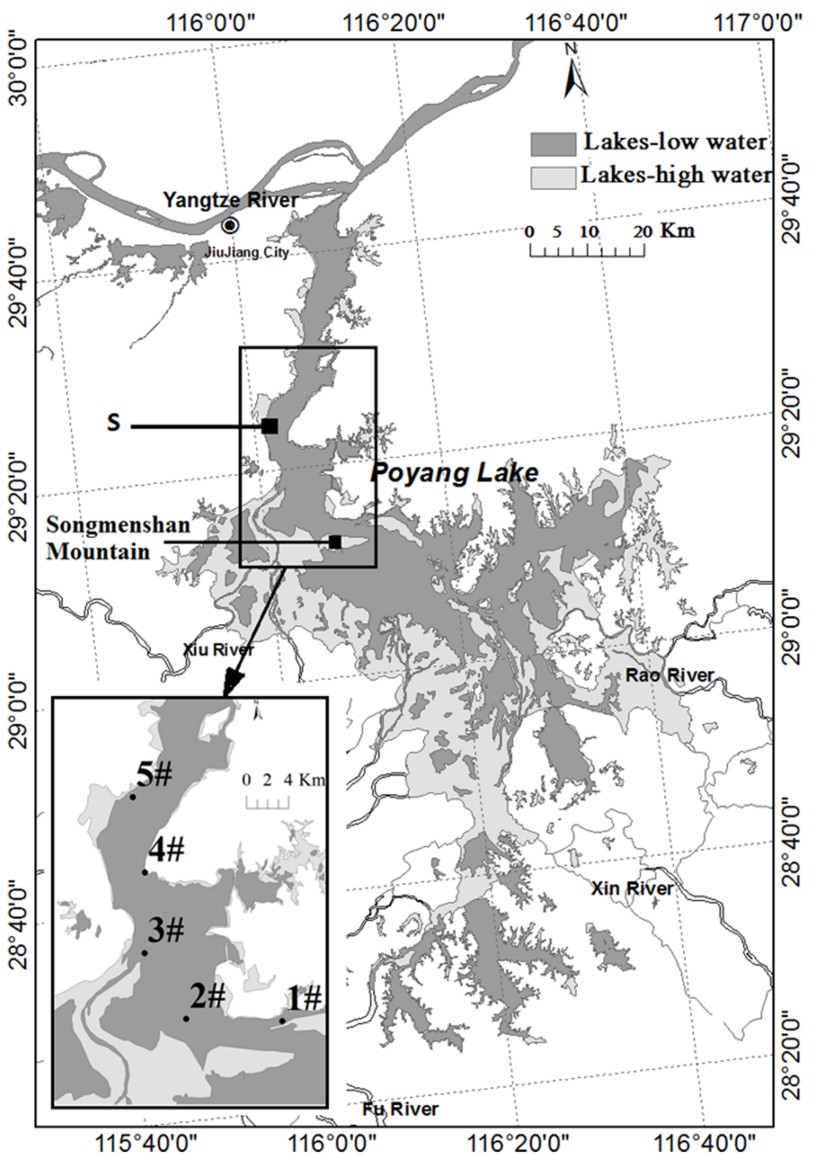

**Figure 1 Map of Poyang Lake indicating Songmen Mountain and the reference point "S" (refer to text).** The insert shows the area at 13 m water level in dark gray and at 20 m water level in light gray. The locations of the sampling stations are indicated by numbers #1–5.

16 and 20 m. It disperses into several smaller water bodies in the low water level phase when the water level decreases to 13 m. It is divided into two parts by Songmenshan Mountain, the point "S" in Fig. 1. The northern part is the water channel joining the Yangtze River, with the length of 40 km and the width of three to five km (the narrowest point is about 2.8 km). The area in the northern part between Duchang and Xingzi counties of Poyang Lake was selected for the study (Fig. 1). This area received all the *Anabaena* from the southern part of the lake. This area forms a narrow meandering channel during low water level phase in winter. Cyanobacteria (mostly *Anabaena*, *Microcystis,* and *Planktothrix*) gathered in this area in summer and autumn.

## Methods

Samples were collected weekly or bi-weekly during the periods May–November in 2013 and June–December in 2014. Five stations were selected to sample (Fig. 1). Water transparency was estimated using a ~30 cm Secchi disc. Water temperature and pH were measured in situ with multiparameter profiler YSI 6600 V2. Water depth was measured using a handheld Speedtech Depthmate portable sounder. The water samples were collected by a "Ruttner"-sampler at three depths (surface, middle, and bottom layers of the lake) and mixed in a clean bucket as the final sample at each station. Phytoplankton sub-samples (1,000 mL) were immediately fixed with 10 mL Lugol's Iodine solution. The phytoplankton taxon was identified and enumerated by the inverted-microscope Nikon TS100-F following the sedimentation and inverted microscope method of *Utermöhl (1958)*. The algal division, taxa, genus, and species were identified according to *Hu & Wei (2006)* and the biomass estimated by volume. Heterocyst frequency, the number of heterocyst per unit length of filament (*Chan et al., 2004*), were used as an indicator of the $N_2$-fixation capacity of the Cyanobacteria (*Laamanen & Kuosa, 2005*). Water chemical variables ($NO_2$-N, $NO_3$-N, $NH_4$-N, and $PO_4$-P) samples were filtered with 0.2 μm syringe filters before determined. Total nitrogen (TN) and total phosphorus (TP) were determined by persulfate oxidation and spectrophotometry (*Jin & Tu, 1990*). Total P was oxidized to $PO_4^{3-}$ at 120 °C. Dissolved inorganic nitrogen (DIN) is the sum of concentrations of $NO_2$-N, $NO_3$-N, and $NH_4$-N. The chlorophyll *a* concentration was determined according to *Lorenzen (1967)*. Water level data for the period January 2013–December 2014 were obtained from the hydrology of the Jiangxi Province website (http://www.jxsl.gov.cn/id_jhsq201404101112508271/column.shtml).

All calculations were completed with the statistical package SPSS for Windows (version 17.0). One-way analysis of variance tests (ANOVA) was used to determine if the mass concentrations of dissolved nutrients in water ($NH_4^+$-N, $NO_3^-$-N, and $PO_4^{3-}$-P) differed among the sampling periods. A Spearman rank correlation test was performed to detect correlations between measured physicochemical parameters and the flooding regime to *Anabaena* heterocyst frequency in Poyang Lake. Graphs were made with SigmaPlot 12.0 (Systat Software, Inc., San Jose, CA, USA).

## RESULTS

Water level changed from 8.67 to 16.70 m in 2013 and from 7.37 to 18.59 m in 2014 (Fig. 2; Table 1). Water temperature ranged from 15 to 33 °C in 2013 and 12 to 30 °C in 2014. The high-water level coincided with the summer temperatures. Water transparency was about 0.5–0.7 m in May and June at the beginning of the increasing water level phase and it decreased to 0.3–0.4 m in August of 2013. Water transparency ranged from 0.25 to 0.5 m in 2014.

The seasonal change in water level triggers variable nutrient concentrations largely due to dilution (Table 1; Fig. 3). Total nitrogen concentrations varied from 0.49 to 3.07 mg L$^{-1}$ in the increasing water level phase, averaging 1.70 mg L$^{-1}$, and ranged from 1.21 to 3.44 mg L$^{-1}$ in the high-water level phase, averaging 1.85 mg L$^{-1}$. Nitrate concentrations were between 0.24 and 1.52 mg L$^{-1}$ in the increasing water level phase, averaging

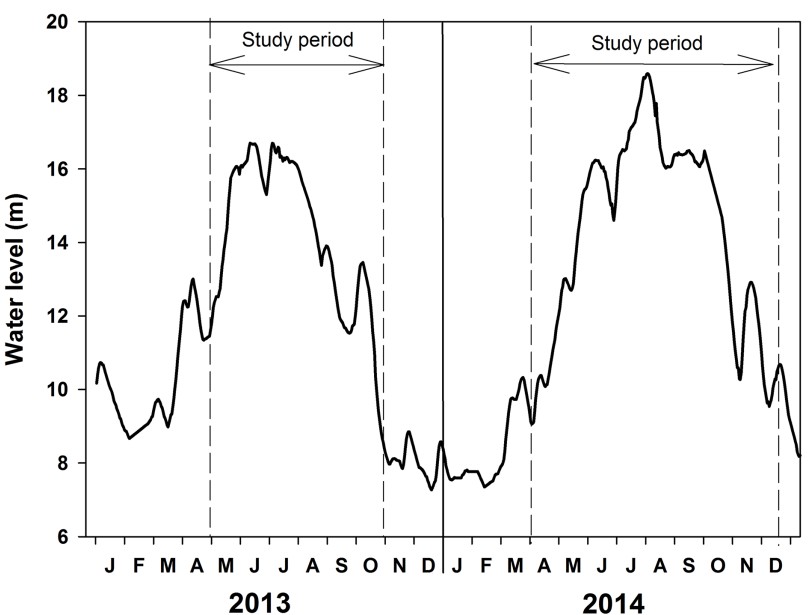

**Figure 2 Water level fluctuations of Poyang Lake in the years 2013 and 2014.**

**Table 1 Summary of statistical data for the measured variables.**

|  | Mean | Max | Min | Std Dev |
|---|---|---|---|---|
| WL (m) | 13.79 | 18.26 | 7.6 | 3.10 |
| WT (°C) | 25.51 | 33.31 | 10.03 | 5.41 |
| Transparency (m) | 0.37 | 1.0 | 0.1 | 0.17 |
| SS (mg L$^{-1}$) | 35.90 | 203.3 | 1.4 | 33.71 |
| Turbidity (NTU) | 45.22 | 418 | 2.4 | 37.84 |
| pH | 8.38 | 10.38 | 6.72 | 0.56 |
| Conductivity (ms cm$^{-1}$) | 130.1 | 495.7 | 37.9 | 49.7 |
| TN (mg L$^{-1}$) | 1.984 | 7.3 | 0.04 | 1.07 |
| NO$_3$-N (mg L$^{-1}$) | 0.99 | 2.17 | 0.07 | 0.34 |
| NO$_2$-N (mg L$^{-1}$) | 0.039 | 0.18 | 0.0001 | 0.026 |
| NH$_4$-N (mg L$^{-1}$) | 0.157 | 1.33 | 0.001 | 0.157 |
| TP (mg L$^{-1}$) | 0.12 | 3.63 | 0.01 | 0.29 |
| PO$_4$-P (mg L$^{-1}$) | 0.05 | 0.85 | 0.001 | 0.067 |
| DIN/DIP | 43.56 | 172.43 | 0.8 | 38.16 |

0.87 mg L$^{-1}$, and varied from 0.66 to 1.63 mg L$^{-1}$ in the high-water level phase, averaging 0.95 mg L$^{-1}$. The range of nitrite concentrations was 0.019–0.053 mg L$^{-1}$ in the increasing water level phase, averaging 0.037 mg L$^{-1}$, and varied from 0.012 to 0.067 mg L$^{-1}$ in the high water level phase, averaging 0.037 mg L$^{-1}$. Ammonium concentrations were between 0.076 and 0.182 mg L$^{-1}$ in the increasing water level phase, averaging 0.129 mg L$^{-1}$, and ranged from 0.016 to 0.161 mg L$^{-1}$ in the high water level phase, averaging 0.089 mg L$^{-1}$. Concentrations of PO$_4$-P were 0.003–0.191 mg L$^{-1}$ in the increasing water level phase, averaging 0.038 mg L$^{-1}$, and varied from 0.011 to 0.217 mg L$^{-1}$ in the high

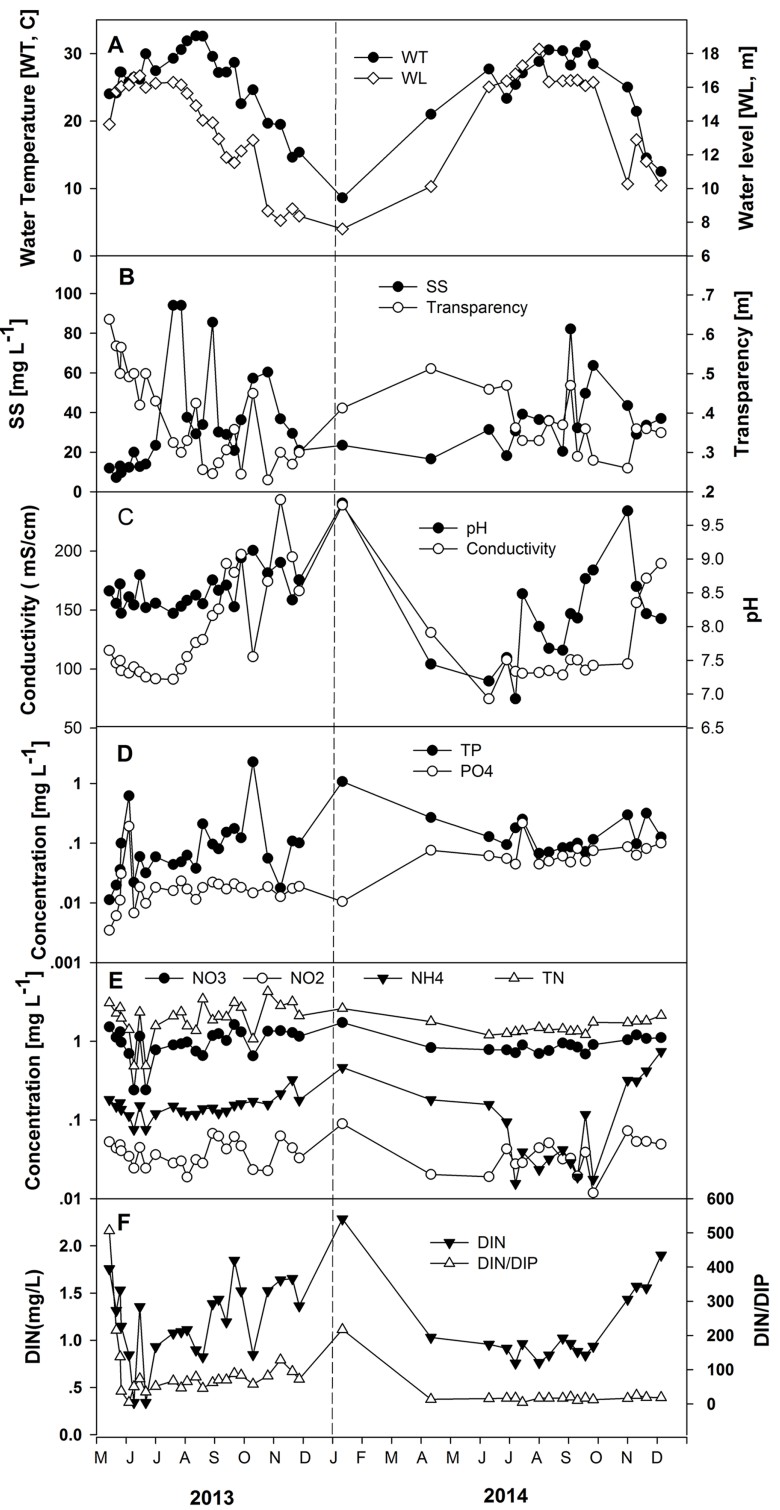

**Figure 3 Variations of the measured environmental parameters in the years 2013 and 2014.**
(A) Water temperature and water level, (B) suspended solids (SS) and transparency, (C) conductivity and pH, (D) total phosphorus (TP-P) and phosphate ($PO_4$-P), (E) total nitrogen (TN-N), nitrite ($NO_2$-N), nitrate ($NO_3$-N) and ammonium ($NH_4$-N), and (F) DIN and DIN/DIP.

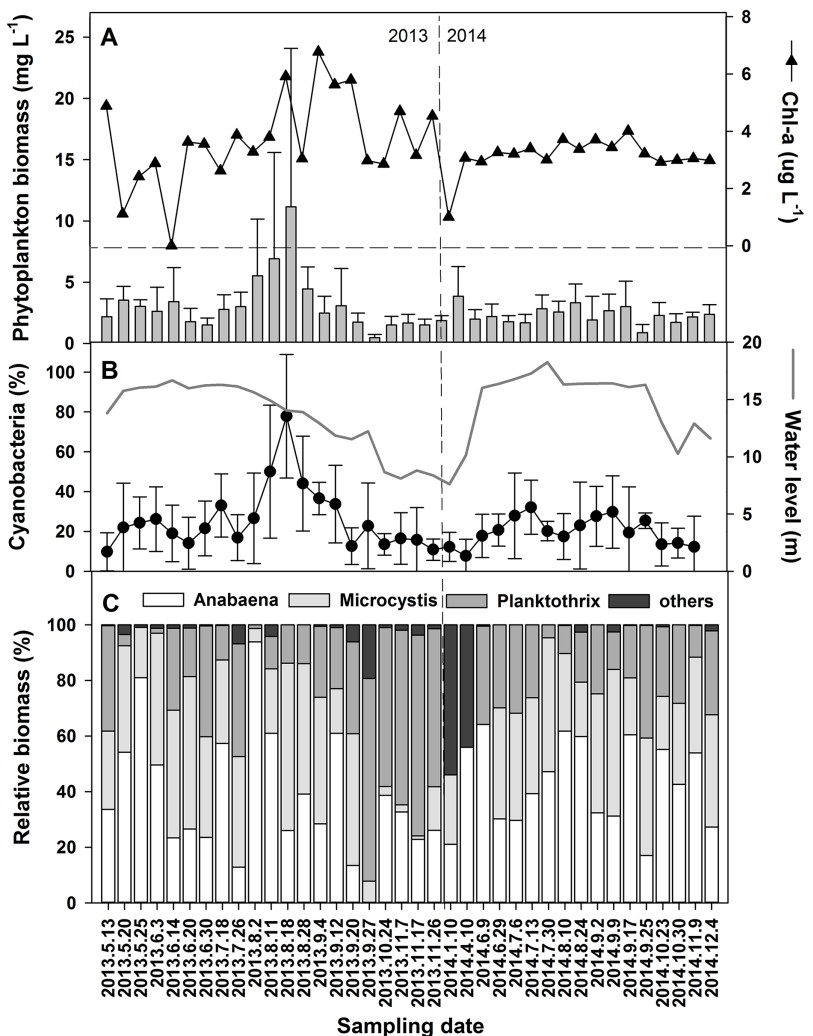

**Figure 4 Variations of the phytoplankton biomass and chorophyll-a in the years 2013 and 2014.** (A) Phytoplankton biomass and chorophyll-a. (B) Percent of Cyanobacteria in the total phytoplankton biomass. (C) The relative biomass of the dominant Cyanobacteria genera. For references, the water level changes from Fig. 2 is inserted as a gray line in (B).

water level phase, averaging 0.045 mg L$^{-1}$ (Fig. 3). Concentrations from 0.011 to 0.621 mg L$^{-1}$ characterized TP in the increasing water level phase, averaging 0.108 mg L$^{-1}$, and were 0.038–0.252 mg L$^{-1}$ in the high-water level phase, averaging 0.108 mg L$^{-1}$. The ratio DIN/DIP lies between 4.41 and 506.78 in the increasing water level phase, averaging 104.02, and was 4.43–88.76 mg L$^{-1}$ in the high-water level phase, averaging 42.73 (Table 1). There is a considerable difference in nutrient concentrations between the increasing and high water level for $NH_4$-N, $NO_2$-N, $PO_4$-P, as well as for transparency based on one-way ANOVA analysis. Values were significantly different ($p < 0.05$) among the two water level phases.

Significant differences in phytoplankton community composition in different hydrological phases were associated with physicochemical variations leading by water level changes. Cyanobacteria contributed 78.5% to the average 11.1 mg L$^{-1}$ of total

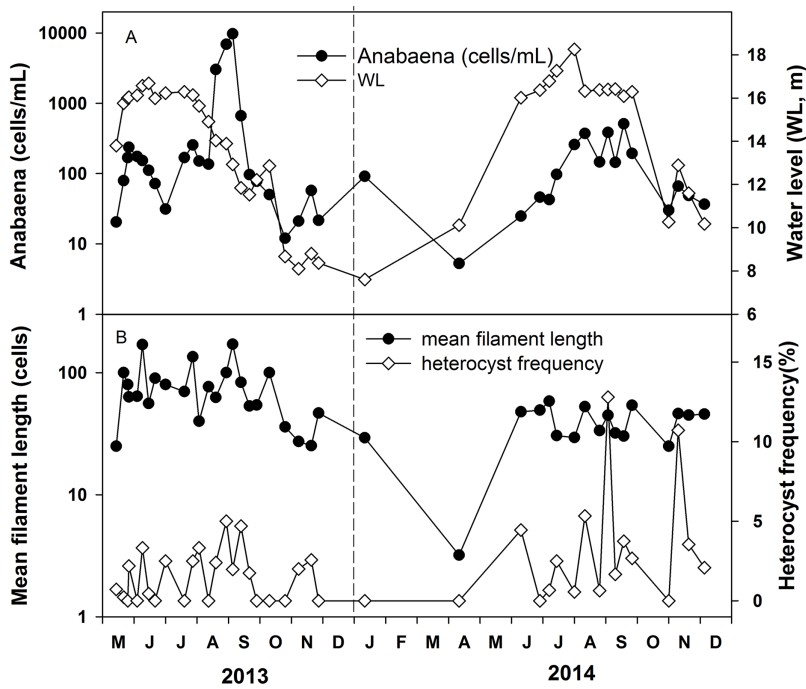

**Figure 5 Variations of Anabaena abundance and Anabaena heterocyst frequency in the years 2013 and 2014.** (A) *Anabaena* abundance. (B) Mean filament length and heterocyst frequency.

phytoplankton biomass in the high-water level phase of 2013 (Figs. 4A and 4B). The dominant genera were *Microcystis*, *Planktothrix*, and *Anabaena* (Fig. 4C). Among the six species of *Anabaena* identified, namely *Anabaena affinis*, *A. planctonica*, *A. smithii*, *A. circinalis*, *A. spiroides*, and *A. azotica*, the last two were the most prominent. The average relative biomass of Anabaena was 43.6% during the study periods. Higher contributions occurred in May, July, and August of 2013 and were over 40% during most of 2014.

Anabaena abundance ranged from 5 to 9,500 cells mL$^{-1}$ (Fig. 5A). It was around 50–150 cells mL$^{-1}$ in most of the time and the high values of 3,000–9,500 cells mL$^{-1}$ were recorded in May and August 2013 and 300–500 cells mL$^{-1}$ in August and September 2014. The mean filament size was 50 cells, ranging from 25 to 171 cells in 2013, while the mean filament size ranged from 25 cells to 58 cells in 2014. The average heterocyst frequency was 3.42% and 9.63% in August of 2013 and 2014, respectively (Fig. 5B). The average heterocyst frequency was 3.65% in the increasing water level phase, and 3.82% in the high water level phase for these 2 years. The greater percentages of heterocysts during high water level are an indirect indication of possible N-fixation, which is supported by the low nitrogen concentration and the low DIN/DIP ratio in Poyang Lake.

The interdependence between all variables and the parameters of *Anabaena* was evaluated by Spearman Rank Correlation (Table 2). The measured physicochemical variables were significantly correlated with the water level in Poyang Lake. Phytoplankton was positively correlated with water temperature and water transparency, and negatively correlated with SS, turbidity, TN and TP. *Anabaena* biomass was positively

**Table 2 Correlation coefficients (Spearman's rho) between environmental variables and parameters of *Anabaena*.**

|  | Herocyst | Anabaena | Cyanobacteria | phytoplankton | Chla | WL |
|---|---|---|---|---|---|---|
| WL | 0.327** | 0.171* | 0.215** | 0.132 | 0.172* | 1.000 |
| WT | 0.167* | 0.315** | 0.384** | 0.292** | 0.409** | 0.539** |
| Transparency | −0.016 | 0.012 | −0.063 | 0.219** | −0.106 | 0.181* |
| SS | −0.019 | 0.005 | 0.021 | −0.259** | 0.048 | −0.068 |
| Turbidity | −0.049 | −0.111 | −0.099 | −0.245** | −0.070 | −0.310** |
| pH | −0.063 | 0.034 | 0.054 | −0.043 | −0.001 | −0.297** |
| Conductivity | −0.187* | −0.193* | −0.137 | −0.145 | −0.174* | −0.658** |
| TN | −0.157* | −0.160* | −0.174* | −0.150* | −0.118 | −0.507** |
| $NH_4$-N | −0.055 | −0.049 | −0.106 | 0.017 | −0.220** | −0.561** |
| $NO_3$-N | −0.188* | −0.132 | −0.128 | −0.120 | −0.175* | −0.476** |
| TP | 0.099 | −0.111 | −0.065 | −0.151* | −0.055 | 0.072 |
| $PO_4$-P | 0.247** | −0.057 | −0.148 | −0.042 | −0.164* | 0.440** |
| DIN/DIP | −0.255** | 0.020 | 0.093 | 0.023 | 0.043 | −0.642** |

**Notes:**
* Correlation is significant at the 0.05 level (two-tailed).
** Correlation is significant at the 0.01 level (two-tailed).

correlated with water temperature and water level, and negatively correlated with TN. Heterocyst frequency of *Anabaena* was positively correlated with water temperature, water level and $PO_4$-P, and negatively with DIN/DIP, $NO_3$-N, and TN.

## DISCUSSION

The water level changes of 8.03 m in 2013 and 11.22 m in 2014 were by far greater than changes at other sites (*Zohary & Ostrovsky, 2011*; *Nõges, Nõges & Laugaste, 2003*; *Casali et al., 2011*). Alterations in water level intensely influence nutrient cycling and mixing processes, leading to variations of the aquatic biota and phytoplankton dynamics (*Zohary & Ostrovsky, 2011*; *Bakker & Hilt, 2016*). The significant correlation between water level and the physicochemical variables indicated that the regular annual extreme water level change of Poyang Lake is the key driver for variations of environmental parameters in this floodplain lake ecosystem (*Liu, Teubner & Chen, 2016b*).

The high concentrations of TN and TP in Poyang Lake, 1.5 and 0.13 mg $L^{-1}$, respectively, was favorable for the propagation and proliferation of algae in the system (*Deng et al., 2011*; *Zhen et al., 2011*). Phytoplankton was positively correlated with water temperature and water transparency, and negatively correlated with SS, turbidity and TN. Phytoplankton was negatively correlated with TN and TP in the present study, indicating transporting of the algae from the southern area to the northern area throughout the water flow (*Qian, Liu & Chen, 2016a*). Cyanobacterial blooms, as the consequence of buoyant migration to the lake surface, were reported by *Liu et al. (2016a)* and *Qian, Liu & Chen (2016a)* from the summer and autumn in 2012 and 2013 in some regions. Cyanobacteria were usually subdominant in Lake Poyang (*Qian, Liu & Chen, 2016a*), and became temporarily the dominant algal group accounting for about 57% of total phytoplankton biomass replacing diatoms in August 2013 in Lake Poyang

(*Qian, Liu & Chen, 2016a*). The Cyanobacterial biomass was significantly greater in the Eastern Bay (lentic region) than in the lotic region of Northern Poyang Lake (*Liu et al., 2016a*). The higher nutrient concentrations and phytoplankton biomass in the northern part of the lake is a result of nutrient discharge from the southern part of the lake (*Liu et al., 2016a*; *Liu, Qian & Chen, 2015*). These results indicate transportation of Cyanobacteria with the water flowing from the southern part to the northern part of Poyang Lake because of the buoyant characteristics of Cyanobacteria (*Liu et al., 2016a*; *Liu, Qian & Chen, 2015*).

*Anabaena* biomass was positively correlated with water temperature and water level, and negatively correlated with TN. Similar as in the floodplain lakes of the Paraná basin during the warm season (*Emiliani, 1990*, *1993*), high biomass of $N_2$-fixing *Anabaena* were documented in Poyang Lake. The peak of *Anabaena* in 2013 was higher than in 2014 because there is a small water level rising in April, which brought the nutrients for the growth of *Anabaena* and also caused the gathering of *Anabaena* in this area. In general, warm water temperatures are necessary for the growth of diazotrophic organisms as *Stal (2009)* pointed out. Consequently, the development of the *Anabaena* population in Poyang Lake was strongly correlated with water temperature. The highest biomass occurred and peaked at times when water temperatures were maximal in summer (*Qian, Liu & Chen, 2016a*), similar to what has been reported elsewhere (*Laamanen & Kuosa, 2005*). However, *Anabaena* can be recorded in the water column of Poyang Lake throughout the year. In essence, *Anabaena* biomass varied synchronously with changes in water temperature.

Cyanobacterial $N_2$ fixation is of ecological importance in aquatic environments (*Karl et al., 2002*). When N/P ratios are low, as is often the case in other floodplain lakes, phytoplankton communities become dominated by heterocyst-bearing Cyanobacteria capable of $N_2$ fixation (*Lewis & Wurtsbaugh, 2008*). Phytoplankton assemblages were dominated by N-fixing Cyanobacteria when N/P ratios were low in mesocosm experiments (*Schindler, 1977*; *Smith, 1983*; *Levine & Schindle, 1999*; *Smith & Bennett, 1999*; *Vrede et al., 2009*). When the N/P ratio dropped in the spring, *Anabaena* rapidly developed and became the dominant biomass component in summer in Chaohu Lake (*Deng et al., 2007*). *Anabaena* is one genus of filamentous Cyanobacteria that can exist as plankton and is known for its nitrogen-fixing abilities. The formation of heterocysts is induced by the lack of combined nitrogen in the medium (*Schindler, 1977*) and the number of heterocysts correlates with the $N_2$ fixation activity of the population (*Lindahl, Wallström & Brattberg, 1980*; *Riddolls, 1985*). When DIN concentration is at limiting levels, the ability to fix $N_2$ is an advantage for Cyanobacteria bearing heterocysts (*Smith, 1983*; *Hense & Beckmann, 2006*; *Piehler et al., 2009*). In the present study, heterocyst frequency of *Anabaena* was positively correlated with water temperature, water level and $PO_4$-P, and negatively with DIN/DIP, $NO_3$-N, and TN, which is the reason for the increase of heterocyst frequency at high water level in both years. Therefore, *Anabaena* was able to propagate with low nitrogen concentration.

In addition, *Anabaena* is buoyant, enabling it to float to the surface to make the most of the available light and atmospheric nitrogen (*Walsby et al., 1989*). This is the reason that

visible floating migration to the lake surface of Cyanobacteria blooms occurred during summer and autumn in 2013–2014 in some regions of Poyang Lake when *Anabaena* was the dominant species (*Qian et al., 2016b*). *Anabaena* can fix atmospheric nitrogen at such low levels of nitrogen in Poyang Lake during the high water level phase that their growth is not limited. *Anabaena* biomass was positively correlated with water temperature and water level, and negatively correlated with TN, which is the reason for the increase of heterocyst frequency in the present study.

Water level fluctuations have an overall impact on phytoplankton community composition, through affecting the physical and chemical variables in Poyang Lake (*Qian, Liu & Chen, 2016a*), which can be verified by the significant correlation between water level and the measured environmental variables. Low and high water level periods differed in several water physicochemical characteristics in Poyang Lake, such as $NO_2^-$-N, $NO_3^-$-N, TN, and TP. The water level change coincided with the season in Poyang Lake. The summer population of *Anabaena* usually contains heterocysts, while filaments were usually devoid of heterocysts in winter and early spring (*Laamanen, 1996*). The average contribution of Anabaena to Cyanobacterial biomass was 43.55% and heterocyst frequency was higher than 3% in high-water level phases in Poyang Lake. We conclude that variations in environmental parameters related to both seasonal variations and water level changes triggered the variations in *Anabaena* biomass and heterocyst frequency.

## CONCLUSIONS

Cyanobacteria biomass accounted for about 57% of total phytoplankton biomass, temporarily becoming the dominant species and even replacing diatoms in August 2013. The dominant species of Cyanobacteria were nitrogen-fixing *Anabaena,* which were present throughout the year. The average relative biomass of *Anabaena* during the study period was over 40%, being most abundant in summer. When the water level increased, *Anabaena* became the dominant species and produced a high number of heterocysts, reflecting relatively nitrogen deficiency in Lake Poyang during this period. The results of the study support the hypothesis that increasing biomass and heterocyst formation of *Anabaena* were primarily triggered by variations in the physicochemical factors, such as high water temperature and suitable nutrients in summer and autumn, driven by the regular annual extreme water level change of Poyang Lake.

## ACKNOWLEDGEMENTS

We want to thank all colleges who collected and processed samples for the monitoring program from the Poyang Lake Laboratory for Wetland Ecosystem Research.

### Funding

This work was financially supported by the National Natural Science Foundation of China (No. 31600345) and Jiangsu Government Scholarship for Overseas Studies (JS-2017-156).

The funders had no role in study design, data collection and analysis, decision to publish, or preparation of the manuscript.

### Grant Disclosures
The following grant information was disclosed by the authors:
National Natural Science Foundation of China: 31600345.
Jiangsu Government Scholarship for Overseas Studies: JS-2017-156.

### Competing Interests
The authors declare that they have no competing interests.

### Author Contributions
- Kuimei Qian conceived and designed the experiments, performed the experiments, authored or reviewed drafts of the paper, approved the final draft.
- Martin Dokulil analyzed the data, prepared figures and/or tables.
- Yuwei Chen contributed reagents/materials/analysis tools.

### Data Availability
The raw data are available in the Supplemental File.

### Supplemental Information
Supplemental information for this article can be found online at http://dx.doi.org/10.7717/peerj.6608#supplemental-information.

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
