# Peer review of "Do the regular annual extreme water level changes affect the seasonal appearance of Anabaena in Poyang Lake?"

_PeerJ, doi:10.7717/peerj.6608_

## Round 0.1 · original submission · Minor Revisions

Water level fluctuation alters the physical, chemical and also the biological variables of Poyang Lake. Understanding the seasonal variations of Anabaena could be benefit by explicitly considering the Water level fluctuation. The study was examined by two experts. Both of them found it is interesting and thus suggested minor revision. Please follow their comments closely.

Reviewer 1 ·

Basic reporting

Manuscripts can be described in a more professional English language. The figures and tables can clearly present the research results.

Experimental design

The experimental design of this research is reasonable and can clearly verify the scientific hypothesis.

Validity of the findings

The scientific problems are clear and it has practical guiding significance for the water resources management of Poyang Lake.

Additional comments

The manuscript can be published after revision.
The author needs to answer a few questions.
(1) Why did you only choose northern part between Duchang and Xingzi counties of Poyang Lake for research? Is there any Anabaena occurrence in other water areas of Poyang Lake?
(2) According to the water level data, the water level in 2013 was significantly lower than that in 2014. Why did the authors not compare the difference between two years of Anabaena? Why is there a difference in the peak of Anabaena occurrence in two years?
(3) We recommend increasing the canonical correlation analysis to determine which important environmental factors have significant impact on the biomass and heterocyst frequency change of Anabaena.
(4) We recommend increasing the results of correlation analysis between water level and other water environmental factors.

Reviewer 2 ·

Basic reporting

'no comment'

Experimental design

'no comment'

Validity of the findings

1) What are the main environmental factors affecting the dynamic changes of phytoplankton community structure? Need to be supplemented in the paper.
2) How does the change of water level affect the community structure of phytoplankton? The changes of biomass, community structure and population dynamics need to be further elaborated.
3) There are several kinds of algae in the composition of cyanobacteria. Is this the change of influence on Anabaena? Is it the same for other cyanobacteria or only for Anabaena?
4) Does the change of water level imply the change of other environmental factors? Additional explanations are needed in this regard.
5) There are many papers on the effect of water level on phytoplankton,
which need to be further supplemented.

Additional comments

The change of water level has an impact on the structure of phytoplankton community. On the one hand, it affects the composition of phytoplankton community, on the other hand, it affects the size of individual phytoplankton. The author studied the relationship between the water level change of Poyang Lake and Anabaena, which has a good role in guiding the management of water resources, but it needs further modification and can be considered for publication.

---

## Round 0.2 · Minor Revisions

I think that the reviewers' comments are well considered and replied. This study shows an important contributions to our understandings of the seasonal variations of Anabaena in Poyang Lake. As such, the article is scientifically acceptable.

However a check by the Section Editors for the journal has shown that the English language needs considerable improvement. Please can we ask you to further edit the language, as PeerJ does not perform language editing as part of the production process.

---

## Round 0.3 · accepted · Accept

Your article is scientifically Acceptable, however a final check of the manuscript by James Reimer, one of the Section Editors for this part of the journal, showed that it still needs another round of English editing. Please can we ask you to further edit the language before we can Accept the manuscript.